# OPF/PMMA Cage System as an Alternative Approach for the Treatment of Vertebral Corpectomy

**Asghar Rezaei [1,2]**, **Hugo Giambini [3]**, **Alan L. Miller II [2]**, **Xifeng Liu [1,2]**,
**Benjamin D. Elder [1,2,4]**, **Michael J. Yaszemski [1,2]** and **Lichun Lu [1,2,*]**

1   Department of Physiology and Biomedical Engineering, Mayo Clinic, Rochester, MN 55905, USA;
    Rezaei.Asghar@mayo.edu (A.R.); Liu.Xifeng@mayo.edu (X.L.); Elder.Benjamin@mayo.edu (B.D.E.);
    Yaszemski.Michael@mayo.edu (M.J.Y.)
2   Department of Orthopaedic Surgery, Mayo Clinic, Rochester, MN 55905, USA; Miller.Alan@mayo.edu
3   Department of Biomedical Engineering and Chemical Engineering, The University of Texas at San Antonio,
    San Antonio, TX 78249, USA; Hugo.Giambini@utsa.edu
4   Department of Neurosurgery, Mayo Clinic, Rochester, MN 55905, USA
*   Correspondence: Lu.Lichun@mayo.edu; Tel.: +1-507-284-2267

**Abstract:** The spinal column is the most common site for bone metastasis. Vertebral metastases with instability have historically been treated with corpectomy of the affected vertebral body and adjacent intervertebral discs, and have more recently been treated with separation surgery. With demographics shifting towards an elderly population, a less-invasive surgical approach is necessary for the repair of vertebral defects. We modified a previously reported expandable hollow cage composed of an oligo[poly(ethylene glycol) fumarate] (OPF) containment system that could be delivered via a posterior-only approach. Then, the polymer of interest, poly (methyl methacrylate) (PMMA) bone cement, was injected into the lumen of the cage after expansion to form an OPF/PMMA cage. We compared six different cage formulations to account for vertebral body and defect size, and performed a cage characterization via expansion kinetics and mechanical testing evaluations. Additionally, we investigated the feasibility of the OPF/PMMA cage in providing spine stability via kinematic analyses. The in-vitro placement of the implant using our OPF/PMMA cage system showed improvement and mechanical stability in a flexion motion. The results demonstrated that the formulation and technique presented in the current study have the potential to improve surgical outcomes in minimally invasive procedures on the spine.

**Keywords:** spine; expandable cage; kinematic testing; minimally invasive surgery; OPF formulation

## 1. Introduction

The spinal column is the most common site for bone metastasis, with the latter occurring in one third of cancers [1–4]. Other pathologies including fractures and infection can also affect the vertebrae, increasing the risk for vertebral collapse or nervous tissue injury, and reducing spinal stability [5]. Vertebral collapse with instability, from tumors, trauma, or infections, has historically been treated with corpectomy of the affected vertebral body and adjacent intervertebral discs. This defect can then be replaced with structural autografts, allografts, or a variety of titanium or polymeric cages [6,7] with the purpose of reducing pain, decompression of neural elements, spinal stabilization, and resection of the malignancy [8]. Even though these materials have specific advantages, they have their own limitations that may result in postsurgical complications and negatively affect the intended outcomes [9–11]. In addition, these open surgical procedures require a significant surgical exposure, typically through a thoracotomy or costotransversectomy, which is highly invasive [12] particularly in elderly and frail

patients, posing greater surgical risks and a challenging recovery [13]. With demographics shifting towards an elderly population, a less-invasive surgical approach is necessary for the repair of vertebral defects so commonly present in this population [14].

To achieve this goal, a polymeric expandable cage composed of oligo[poly(ethylene glycol) fumarate] (OPF) was previously developed that could be delivered via a posterior-only surgical approach [15]. The OPF hollow cage can expand to a predetermined size within a surgical time frame making it possible to perform a less-invasive surgery compared with current surgical approaches, and minimizing associated complications with the procedure.

An important limitation of our previous study was the size of the hollow cage, which did not conform to in-vivo measurements of a defect. Thus, in the current study, we modified our polymeric formulations to allow for a more accurate cage size. We then performed expansion kinetics, augmented the cages with poly(methyl methacrylate) (PMMA) to form an OPF/PMMA cage for spine stability, and conducted kinematic analysis on a cadaveric spine as a proof-of-concept to evaluate the range of motion (ROM) and the effect of corpectomy and treatment. Therefore, the purpose of the current study was twofold: first, to optimize the cage formulations to account for vertebral body and intervertebral discs' defect size, and perform a characterization via expansion kinetics and mechanical testing evaluations; second, to investigate the feasibility of the OPF/PMMA cage in providing spine stability via kinematic analyses.

## 2. Materials and Methods

### 2.1. OPF Expandable Cage

#### 2.1.1. OPF Synthesis

OPF was synthesized using fumaryl chloride (Sigma Aldrich Co., Milwaukee, WI, USA) and poly(ethylene glycol) (PEG; Sigma Aldrich Co., Milwaukee, WI, USA) with an average molecular weight of 2000 Da, as previously described [15]. Briefly, PEG (100 g) was placed in a two-neck flask in an ice bath and purged with nitrogen for 10 min; this process was repeated three times. Then, 1000 mL of anhydrous methylene chloride ($CH_2Cl_2$, Fisher, Pittsburgh, PA, USA) and molecular sieves (3Å, beads 4-8 mesh; Sigma Aldrich Co., Milwaukee, WI, USA) were added to the flask to dissolve the PEG and remove any water. Potassium carbonate powder ($K_2CO_3$; Sigma Aldrich Co., Milwaukee, WI, USA) (40 g) was then added, with a subsequent dropwise addition of fumaryl chloride (1:1 in molar ratio to PEG) under stirring conditions. The reaction was kept at room temperature for 48 h and then filtered to remove the solid $K_2CO_3$ powder. The filtrate was concentrated by rotary evaporation to remove any $CH_2Cl_2$ remnant and precipitated in 1 L diethyl-ether at −20 °C overnight. The precipitate was then filtered and fully dried in vacuum. Before storing the synthesized OPF at −20 °C for future use, gel permeation chromatography (GPC) was performed to evaluate its molecular weight. Briefly, 34 mg of OPF was dissolved in 3.5 g of tetrahydrofuran (THF) and the solution was run four times to obtain an average molecular weight.

#### 2.1.2. Fabrication and Expansion of Cages

The synthesized OPF of molecular weight (Mn) of ~4000 Da was used to fabricate cylindrical hollow cages of 12 mm in diameter. Six different formulations were used to improve the rigidity and failure of the hollow cages upon expansion (Table 1). 1-Vinyl-2-pyrrolidinone (NVP), as a crosslinker, was provided from Sigma Aldrich Co., Milwaukee, WI, USA.

**Table 1.** Six different formulations implemented to make oligo[poly(ethylene glycol) fumarate] (OPF) hollow cages. NVP: 1-Vinyl-2-pyrrolidinone.

| Formulations | OPF (g) | NVP (mL) | BAPO (g) | $CH_2Cl_2$ (mL) |
|:---:|:---:|:---:|:---:|:---:|
| a | 1 | 0.1 | 0.05 | 2 |
| b | 1 | 0.01 | 0.05 | 2 |
| c | 1 | 0.001 | 0.05 | 2 |
| d | 1 | 0.01 | 0.1 | 2 |
| e | 1 | 0.001 | 0.1 | 2 |
| f | 1 | 0 | 0.05 | 2 |

For each formulation, OPF, phenyl bis (2,4,6-trimethylbenzoyl)-phosphine oxide (BAPO) as a photo initiator, and $CH_2Cl_2$ were combined into a flask according to Table 1. The mixture was vortexed until all solutes were dissolved. As a cylindrical mold, a metal rod of 12 mm in diameter was fabricated to construct the inner diameter of the OPF hollow cage. After greasing the metal rod, the resin was added to the mold. The mold containing the resin was placed in a UV oven to initiate crosslinking. After 1 h, the center metal rod of the mold was removed and the hollow cage placed back in the UV oven for an additional 2 h. The cage was placed in the fume hood at room temperature overnight to fully dry.

Sodium methacrylate (SMA) was then physically infiltrated into the network for faster expansions of the OPF hollow cages. The crosslinked OPF cages were soaked for 8 h in ddH2O containing 0.5% SMA. Cages were subsequently dried overnight in a 70 °C oven by placing the expanded hollow cages around the metal mold. Because drying of the 12-mm-diameter cages in the oven induced shrinking, the cages were dried on an 8-mm, metal-rod mold, leading to a thicker, cylindrical, and uniform cage (Figure 1). Expansion kinetics of the cages was then evaluated by immersion in 50 mL of phosphate buffered saline (PBS) at 37 °C. Mass, length, and diameter were measured at serial time points from 0–100 min. The averaged sample dimensions were as follows: for pre-expansion, internal diameter was 8.07 mm, wall thickness was 0.78 mm, and length was 10 mm. After expansion (after 100 min), the internal diameter increased to 17.63 mm, the wall thickness changed to 1.74 mm, and the length increased to 21.39 mm.

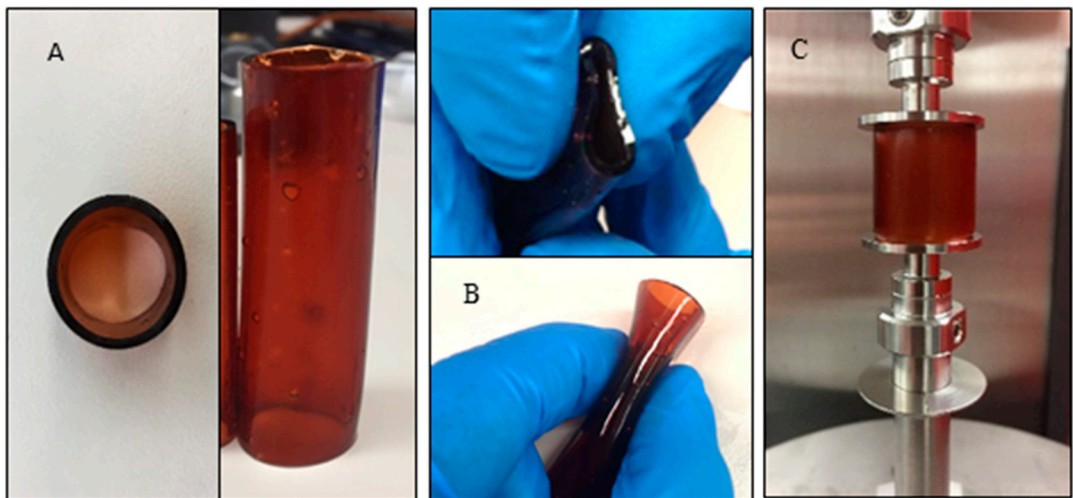

**Figure 1.** The 12 mm OPF hollow cage: (**A**) top and side views, showing a uniform geometry; (**B**) the flexibility of the cages for placement into a surgical site; and (**C**) mechanical testing setup.

### 2.1.3. Mechanical Testing

Mechanical testing was performed on the OPF hollow cages before and after expansion using a dynamic mechanical analyzer (DMA) (RSA-G2, TA instruments, New Castle, DE, USA) (Figure 1C).

Three specimens were tested in each group. Compressive stress–strain curves for the OPF cages were obtained at a constant linear rate of 0.01 mm/s. Compressive moduli were calculated from the slope of the linear region in the stress–strain curves and averaged. As a proof-of-concept and before evaluating the response of the cages in our cadaveric testing, poly(methyl methacrylate) (PMMA) was used to form OPF/PMMA cages. A mold, comprised of two rods, was used to mimic the superior and inferior vertebral bodies, and the cage was placed in between the rods. Using a syringe, the hollow cage was filled with PMMA to improve the overall rigidity and stability of the construct.

*2.2. Kinematic Motion Testing on Spine*

The testing protocol was approved by the Mayo Clinic institutional review board (IRB) (ID: 16-004936). One cadaveric torso was obtained from the anatomy department in our institution. Screening radiographs were used to ensure no fractures, implants, or significant anatomical anomaly was present in the spine. The lumbar region was chosen as this region has the largest vertebrae in the spinal column and frequently undergoes minimally invasive surgeries such as corpectomy. The lumbar spine ($L_1$-Sacrum) was dissected from the torso leaving the ligaments and muscles surrounding the spine intact. The specimen was then kept frozen at $-20$ °C and thawed overnight at room temperature (65 F) in preparation testing. PMMA resin, k-wires, and screws were used to embed the uppermost vertebral body ($L_1$) and sacrum in circular acrylic fixtures. The potted specimen was then attached to a custom spine simulator which has been previously designed and implemented on various studies [16–18]. The spine simulator device has two passive axes of translation in the transverse plane and a third distal axial stage under load control using a pneumatic system. Stepper motors can generate motions in flexion/extension, lateral bending, and axial rotation, and forces and moments are measured using a 6-component load cell (JR3, Woodland, CA, USA) [16].

Three-dimensional kinematic measurements were obtained using an Optotrak Certus optoelectric data acquisition system (Northern Digital Inc., Waterloo, ON, Canada) with a 3-camera unit and accompanying software (MotionMonitor, Innovative Sports Training Inc., Chicago, IL, USA). Before testing, active motion markers were attached to the $L_2$, $L_4$, $L_5$, and sacrum levels of the spine. The $L_3$ vertebra was left markerless as this is the level that was removed to mimic corpectomy. Fiberglass pins were inserted into each vertebral body and Optotrak sensors were rigidly secured to the pins with aluminum connecting elements. These markers allowed for motion capture of each level as the spine was tested. After marker placement, the spine was placed in the spine simulator and tested in flexion/extension and right/left lateral bending with a pure moment of 7.5 Nm, as previously described [16]. A follower load was not implemented in this study. A follower load is a load that is added to the testing equipment by means of cables to mimic the weight of the torso. Sensor position and moment data from the load cell were synchronized and simultaneously recorded using the MotionMonitor software. Load and displacement data were collected for a total of 4 motion cycles. Spine kinematics for flexion/extension and left/right lateral bending were obtained in the following sequence: (1) intact spine; (2) intact spine with bilateral pedicle screws and instrumented rods at the $L_2$ and $L_4$ vertebral levels; (3) corpectomy of the $L_3$ vertebral body with bilateral pedicle screws and rods at the $L_2$ and $L_4$ vertebral levels; (4) corpectomy of the $L_3$ vertebral body with bilateral pedicle screws and rods at the $L_2$ and $L_4$ vertebral levels, and OPF/PMMA cage. Conditions 1, 3, and 4 are shown in Figure 2. The total angular ROMs (flexion/extension and left/right lateral bending) were determined for the $L_2/L_4$ spine segment, to locally analyze the motion of the segment affected by the OPF/PMMA cage; and for the $L_2$/Sacrum to analyze the global motion of the spine. Custom-made Matlab codes were developed to obtain outcome data for each motion, each level, and each condition.

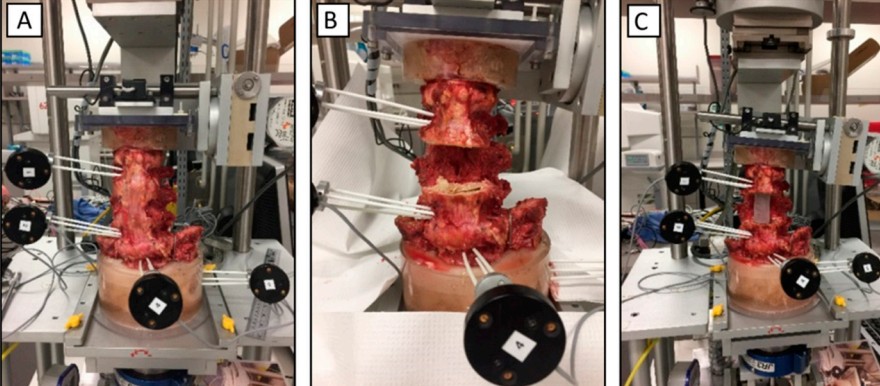

**Figure 2.** Kinematic motion testing setup; (**A**) intact spine; (**B**) spine after corpectomy; and (**C**) spine after placing the poly (methyl methacrylate) (PMMA) rod between $L_2$ and $L_4$.

## 3. Results

### 3.1. Cages Testing

The material of choice for the cage should be able to tolerate significant deformation so that it can be inserted at the surgical site through a small incision. Therefore, as we were developing the formulations and cages, we were qualitatively assessing the malleability and rigidity of the cages. Before proceeding with experimental testing of the cages and in-vitro specimen kinematics, the cage formulations, as shown in Table 1, were compared based on the rigidity of the cages upon manual manipulation and integrity after dehydration in the oven. Formulation "f" showed the best results for the structural integrity of the cage and, thus, was chosen for the study and next steps. The cages made of the remaining formulations broke upon dehydration or were too malleable to serve as a hollow cage for our purpose.

The hollow cage was malleable for easy insertion via a posterior-only approach and could tolerate a considerable amount of deformation without failure (Figure 1B). The ratios of length, wall thickness, and internal diameter were measured for 100 min as shown in Figure 3A. During swelling, the major increase in all the outcomes was found during the first 20 min, in which all of these parameters, including wall thickness, internal diameter, and length, increased about twice their original sizes. No significantly visual changes were seen after 40 min of swelling. The internal diameter increased by up to 2.2-times the original diameter, leading to a final diameter of about 17.6 mm.

Mechanical testing was successfully performed on the cage specimens before and after expansion and the results are shown in Figure 3B,C. After expansion, the cages became very soft and flexible; the stiffness of the cages dropped drastically from 110 N/mm to 0.25 N/mm after swelling (Figure 3B). Similar outcomes were observed in the elastic moduli of the unexpanded and expanded hollow cages, about 54 MPa, and 0.05 MPa, respectively (Figure 3C).

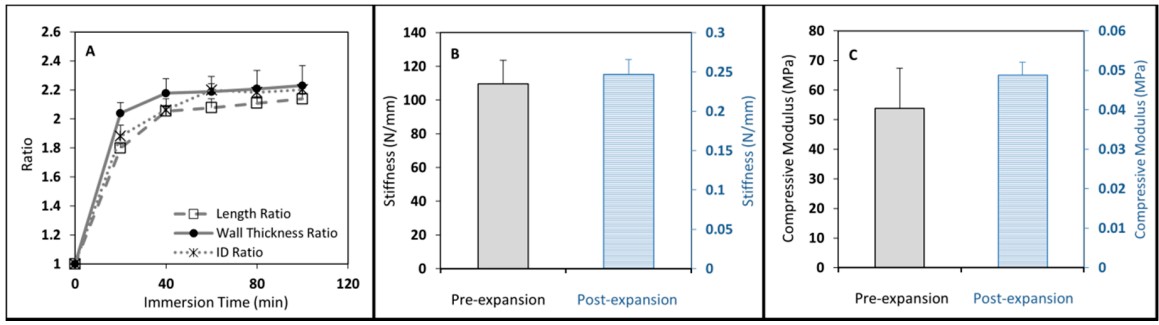

**Figure 3.** (**A–C**): Unexpanded (dry) and expanded 12 mm cages were compressed to obtain compressive mechanical properties. As expected, dry cages presented much higher compressive properties.

### 3.2. Cadaveric Testing

Figure 4 illustrates the bending angle outcomes vs. the applied moments for two loading conditions, flexion/extension and right/left lateral bending. Data were recorded for all four scenarios including a normal/intact spine, normal with posterior fixation, corpectomy with posterior fixation, and fixation with corpectomy plus cage.

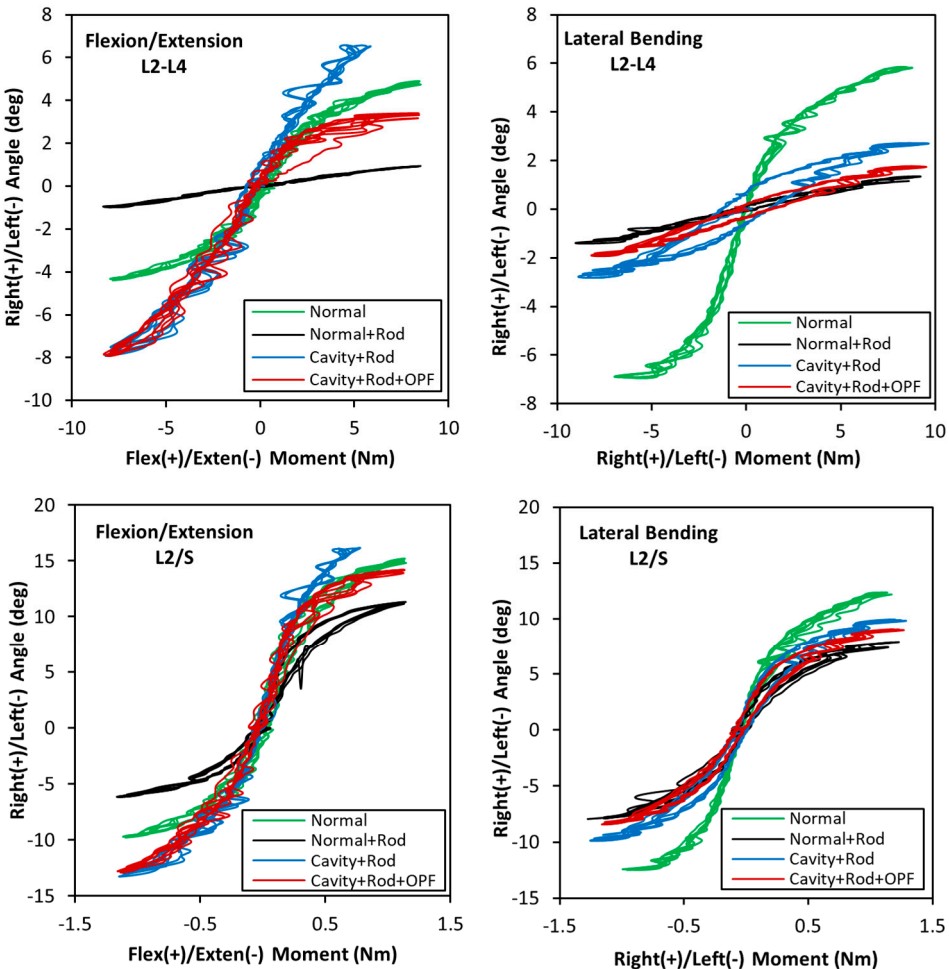

**Figure 4.** Flexion/extension and left/right lateral bending ranges of motion (ROM) for $L_2/L_4$ and $L_2/S$.

The condition of the spine resulted in different kinematic outcomes. In the intact spine (Normal), the $L_2/L_4$ segment moved from $-4.3°$ in extension to $+4.7°$ in flexion. Upon fusing the spine with instrumented rods and screws (Normal + Rod), these motions were greatly reduced to less than $1°$ showing the significant effect of rods and fusion. After performing corpectomy of the vertebral body and adjacent discs (Cavity + Rod), the flexion/extension motions were significantly increased by up to 82%, to about $-8°$ to $+7°$. Finally, the OPF hydrogel was placed between the $L_2$ and $L_4$ (Cavity + Rod + OPF) segment and PMMA injected in the lumen to form the OPF/PMMA cage. During flexion (when the superior vertebra moves anteriorly), the $L_2/L_4$ segment moved $+3.2°$, which was only 32% lower than that of the intact spine, and considerably less than the posteriorly stabilized spine with an unrepaired corpectomy defect. In extension (when the superior vertebra moves posteriorly and tends to separate from the scaffold system), however, the OPF/PMMA cage did not help improve the spine motion as it was not attached to the top and bottom vertebral bodies. This can be noticed in the graph by looking at the extension profiles of the Cavity + Rod + OPF and Cavity + Rod which show similar behaviors. In lateral bending, the $L_2$-$L_4$ segment of the intact spine moved $-6.9°$ to the left and $+5.8°$ to the right. However, the remaining scenarios provided a much smaller ROM from $-2.8°$ to $+2.6°$, stiffening the

segment mainly due to pedicles and screws. It is important to note that in this motion the effect of the cage is minimum as there is no superior/inferior attachment to the endplates. The decrease in motion compared to the Cavity + Rod could be attributed to the friction of the PMMA with the endplates.

The motions of the $L_2/S$ segment are also shown in Figure 4. Compared with the $L_2/L_4$ outcomes, all motions were considerably larger, for example, the intact spine ranged from −13° in flexion to 15.5° in extension. All remaining scenarios showed a similar pattern to that observed in the intact spine. The lateral bending for the intact spine varied from −12.4° to +12°, with the remaining scenarios showing lower and similar outcomes with each other. The $L_2/S$ analyses, showing larger ranges of motion compared to the $L_2/L_4$ segment, are expected, as in the former case there are additional segments which are not fused, increasing the overall ROM of the construct.

## 4. Discussion

In the current study, we presented six different formulations for OPF hydrogel formulations and found a formulation that can be tailored to a specific size, mimicking human vertebral sizes. The OPF hollow cages expanded more than twice their original size during an acceptable surgical time-frame, becoming malleable and allowing for insertion inside a surgery site in a less invasive manner. Additionally, we performed kinematic testing and showed that the OPF/PMMA cage could provide mechanical stability to the spine while allowing for similar motions as the intact spine during flexion.

Spinal cage materials typically range widely, from structural allografts and autografts, pure titanium and its alloys, to ceramics, and plastic such as polyetheretherketone (PEEK). In addition to structural bone grafts, the most commonly used materials are titanium alloys due to their high fracture resistance and biocompatibility. PEEK or PEEK-carbon fiber is also popular as their elastic modulus compares similarly to bone tissue [19]. However, these materials are nondegradable, leading to long-term foreign implant reactions and complications including needing a second intervention for cage removal with pseudarthrosis or cage migration [20]. Our proposed OPF hollow cage, in contrast, is a biocompatible biomaterial that will degrade over time, which makes it an attractive material for spine fusion cages [21–24].

Currently, minimally invasive procedures are being used effectively to treat a wide range of degenerative conditions of the spine. These spinal procedures have led to less pain, less blood loss, a decrease in hospital stays, as well as improving wound healing and allowing a faster return to activities of daily living [25]. While structural cages can be placed into corpectomy defects via both anterior and posterior approaches, these approaches typically require a large exposure which can be highly morbid. For instance, a posterior approach for a cage placement requires a lateral extracavitary or costotransversectomy approach, in which the medial aspect of the rib is removed and often a thoracic nerve root is sacrificed. While titanium and PEEK expandable cages have been used more recently [7], they still require a fairly substantial exposure and have a significant risk of cage subsidence, especially in patients with metastatic tumors in the spine and osteoporosis. In the current study, a small posterior-only incision could be made to place the OPF hollow cage while letting it expand to occupy the lesion and space created during the corpectomy process, with percutaneous placement of pedicle screws for added stability. Finally, a polymer, similar to the PMMA currently used clinically, can be injected into the lumen of the cage to allow for a more rigid construct that can provide rigidity and stability to the spine. The technique and process presented in this study allows for the use of different sizes of cages based on patients' needs and vertebral levels to be treated. This posterior-only approach has the potential to provide substantial improvements on surgical outcomes. In this technique, pre-expanded cages can be fabricated and stored in advance of surgeries. After incising and accessing a surgical site, the cage can be inserted. As soon as it is exposed to the body's fluid, the cage starts expanding to more-than-twice its original size in a timely manner.

To analyze the effects of the corpectomy, pedicle-screw fixation, and the placement of the polymer on the ROMs, we measured the local motion of the treated segment ($L_2/L_4$) as well as the global

behavior of the spine. Importantly, both analyses showed that the cage restricted motion during flexion to almost similar values as an intact spine.

There are several limitations in this study. First, the study was limited to one cadaveric spine; however, the intent of the process was to show a proof-of-concept and describe a technique with the potential for improving surgical outcomes in the treatment of metastasis lesions. Second, different sizes of cages were not investigated. Preliminary studies included the development of 8-mm cages; however, these were not included in the current study as these cages were found to be uneven in shape with varying thicknesses and breakage upon the addition of negative charge and drying in the oven. These problems were caused by drying the 8-mm cage on the 8-mm mold, preventing an increase in wall thickness. Third, the cage did not include any teeth-like structures that could attach to the vertebral endplates. This limitation affected ROM outcomes, especially in extension where the cage remained in place between the superior and inferior vertebrae without imposing a restriction on the motion. While this is of importance, the rigidity of the construct is given by the injected PMMA which solidifies within the lumen of the cage and cannot form into a predefined shape containing teeth. This important shortcoming deserves to be further investigated in future studies. Additionally, instead of an in-situ placement of the cage in the current work, a minimally invasive procedure should be performed in-vitro on cadaveric spines to mimic surgery and insertion of the OPF hydrogel cage, and evaluate its expansion kinetics as well as injection of PMMA. Lastly, while PMMA was employed in this study, it would not lead to fusion of the segment clinically. While this is typically unimportant in patients with metastatic spine tumors with limited life expectancy, for the treatment of defects from trauma or infections, future work should focus on replacing the PMMA with a structural polymer that will allow for spinal fusion. Future studies also should focus on the effect of size and shape of the cage on the spine stability.

In conclusion, the new flexible, expandable OPF formulation presented here allows the development of patient-specific and sizable implants that could be placed at a surgical site using a smaller incision compared with current minimally invasive procedures. Kinematic testing also showed the effectiveness of this technique in providing mechanical stability to the spine. The results suggest that this technique could improve spinal surgical outcomes.

**Author Contributions:** A.R., L.L., H.G., and M.J.Y. collaborated to design the study. Biomaterials were synthesized by H.G., A.L.M.II, A.R., and X.L. Specimen preparation, kinematic testing, and data analysis were performed by H.G., and A.R. This manuscript was written and edited by A.R., H.G., L.L., and B.D.E.—L.L. was responsible for the final approval. All authors have read and agreed to the published version of the manuscript.

**Funding:** This work was supported by an NIH grant from the National Institute of Arthritis and Musculoskeletal and Skin Diseases (R01-AR56212) and an award from the National Institute of Biomedical Imaging and Bioengineering (F32-EB023723).

**Conflicts of Interest:** The authors declare no conflict of interest.

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
