# Peer review of "OPF/PMMA Cage System as an Alternative Approach for the Treatment of Vertebral Corpectomy"

_applsci, doi:10.3390/app10196912_

Round 1

Reviewer 1 Report

To me, the paper is good as it is. I commend the authors for their good work.

Author Response

Thank you for your time and commending our study.

Reviewer 2 Report

Authors claimed that a modified expandable hollow cage composed of an oligo[poly(ethylene glycol) fumarate] (OPF) containment system that could be delivered via a posterior-only approach. The poly(methyl methacrylate)  (PMMA) bone cement, was injected into the lumen of the cage after expansion to form an OPF/PMMA cage. Kinematic testing showed the effectiveness of this technique in providing stability to the spine.

Authors conclude that this new expandable OPF formulation allows developing patient-specific and sizable implants that can minimize surgical incision at a lesion site compared with current minimally invasive procedures. The results suggest that this technique could improve spine surgical outcomes.

Strength: The method is clear and approachable. The results fit the hypothesis. The issue is demanded.

Weakness: The results showed an increased stability at surgical sites; however, were lack of mechanisms. Also, the novelty of OPF/PMMA should be more addressed. 

Suggestion:

The histology analysis may help to prove the mechanisms. 

More references to prove the novelty of this material/technique are required. The references are old. They should include more within 5 years.

Author Response

Authors claimed that a modified expandable hollow cage composed of an oligo[poly(ethylene glycol) fumarate] (OPF) containment system that could be delivered via a posterior-only approach. The poly(methyl methacrylate)  (PMMA) bone cement, was injected into the lumen of the cage after expansion to form an OPF/PMMA cage. Kinematic testing showed the effectiveness of this technique in providing stability to the spine.

Authors conclude that this new expandable OPF formulation allows developing patient-specific and sizable implants that can minimize surgical incision at a lesion site compared with current minimally invasive procedures. The results suggest that this technique could improve spine surgical outcomes.

Strength: The method is clear and approachable. The results fit the hypothesis. The issue is demanded.

Weakness: The results showed an increased stability at surgical sites; however, were lack of mechanisms. Also, the novelty of OPF/PMMA should be more addressed. 

Thank you for the comments! We modified the first paragraph of the discussion to address the novelty of the current study. The text (in page 11) now reads:

“In the current study, we presented six different formulations for OPF hydrogel formulations and found a formulation that can be tailored to a specific size mimicking human vertebral sizes. The OPF hollow cages expanded more than twice their original size during an acceptable surgical time-frame, becoming malleable and allowing for insertion inside a surgery site in a less invasive manner. Additionally, we performed kinematic testing and showed that the OPF/PMMA cage could provide stability to the spine while allowing for similar motions as the intact during flexion.”

Suggestion:

The histology analysis may help to prove the mechanisms. 

Thank you for your suggestion. In the current study, we were investigating macroscale mechanical characteristics of the OPF cage so we did not include histological analysis. Unfortunately, we are unable, at this stage, to add histology analysis to the current paper as the cadavers and cages are no longer available to us. However, we acknowledge that future studies should include these types of analyses.

More references to prove the novelty of this material/technique are required. The references are old. They should include more within 5 years.

Thank you for this comment. We have added more and newer references to address the comment:

  1. Shin H, Temenoff JS, Mikos AG (2003) In vitro cytotoxicity of unsaturated oligo [poly (ethylene glycol) fumarate] macromers and their cross-linked hydrogels. Biomacromolecules 4 (3):552-560
  2. Lam J, Clark EC, Fong EL, Lee EJ, Lu S, Tabata Y, Mikos AG (2016) Data describing the swelling behavior and cytocompatibility of biodegradable polyelectrolyte hydrogels incorporating poly (L-lysine) for applications in cartilage tissue engineering. Data in brief 7:614-619
  3. Piantanida E, Alonci G, Bertucci A, De Cola L (2019) Design of nanocomposite injectable hydrogels for minimally invasive surgery. Accounts of chemical research 52 (8):2101-2112
  4. Gaihre B, Liu X, Lee Miller A, Yaszemski M, Lu L (2020) Poly (Caprolactone Fumarate) and Oligo [Poly (Ethylene Glycol) Fumarate]: Two Decades of Exploration in Biomedical Applications. Polymer Reviews:1-38

Reviewer 3 Report

Dear Authors:

Thank you for submitting the manuscript entitled "OPF/PMMA cage system as an alternative approach 2 for the treatment of vertebral corpectomy". It is an interesting topic. The reviewer appreciates the authors' attempt to optimize the OPF/PMMA cage formulations and to investigate the feasibility in providing spine stability via kinematic analyses. I have the following detailed comments.

1. Introduction
- The introduction is OK.

2. Methods
- Line 79: How were the parameters in these 6 formulations decided?
- Line 102: What was the height of the sample?
- Line 114: In cadaveric testing, why L1-Sacrum were chosen?
- Line 133: Please explain what is a "follower load"?
- Looks like the PMMA rod was not connected to L2 and L4. How did you determine if the cage is providing enough support or not?
- what's the statistic method used?
- In the abstract, the authors stated that the study "optimized the cage formulations", what was done regarding the optimization?

3. Results
- Line 152: Any quantitative comparison between the 6 formulations?
- Line 157: Length, wall thickness, and internal diameter, all of them increased after expansion?
- Figure 4: Are the curves from one specimen or averaged results?
- As the authors mentioned, stiffening in lateral bending was mainly due to pedicles and screws. This again goes back to the question that how did authors determine if the cage is providing enough support or not if both ends were not attached to the vertebrates?

4. Discussion
- What is the purpose of the mechanical testing on the OPF hollow cages before and after expansion? The stiffness has differences in magnitude. What was the clinical relevance of the test result?
- As the author mentioned, one advantage of the OPF hollow cages is that it can expand more than twice their original size during an acceptable surgical time-frame. But as described in Line 190, the procedure seems not likely to be done in a simple and fast way. Any comments?
- Line 202: Please discuss what's new and unique in this study, Please summarize your most important finding in the first paragraph.
- Line 228: How was it controlled to expand the cage to occupy the lesion and space created during the corpectomy?
- Line 246: The statement sounds contradicting itself. If the authors believe the cage having a significant impact in flexion and lateral bending, why would the overall behavior and ROM outcomes still have a similar trend?

Author Response

Thank you for submitting the manuscript entitled "OPF/PMMA cage system as an alternative approach 2 for the treatment of vertebral corpectomy". It is an interesting topic. The reviewer appreciates the authors' attempt to optimize the OPF/PMMA cage formulations and to investigate the feasibility in providing spine stability via kinematic analyses. I have the following detailed comments.

  1. Introduction
    - The introduction is OK.
  2. Methods
    - Line 79: How were the parameters in these 6 formulations decided?

Our previous study [1] was used as the basis to modify the concentrations and formulations. As we were developing these cages for minimal surgeries on the spine, the material of choice should be able to tolerate significant deformation so that it can be inserted at the surgical site through a small incision. Therefore, as we were developing the formulations and cages, we were qualitatively assessing the malleability and rigidity of the cages. This process resulted in formulation ‘f’ as the material of interest that could be used to develop a cage, undergo oven treatment (part of the cage development process) and presented significant amount of deformation without any fracture.

  1. Liu X, Paulsen A, Giambini H, Guo J, Miller AL, Lin P-C, Yaszemski MJ, Lu L (2017) A New Vertebral Body Replacement Strategy Using Expandable Polymeric Cages. Tissue Engineering Part A 23 (5-6):223-232

We added the following sentences to the beginning of the results section (page 7):

“The material of choice for the cage should be able to tolerate significant deformation so that it can be inserted at the surgical site through a small incision. Therefore, as we were developing the formulations and cages, we were qualitatively assessing the malleability and rigidity of the cages.”

- Line 102: What was the height of the sample?

Thank you for the comment. We have added the following to the text to address this comment. The text now reads (page 5):

“The averaged sample dimensions were as follows: for pre-expansion, internal diameter was 8.07mm, wall thickness was 0.78mm and length was 10mm. After expansion (after 100min), the internal diameter increased to 17.63mm, the wall thickness changed to 1.74mm, and in the length increased to 21.39mm length.”

- Line 114: In cadaveric testing, why L1-Sacrum were chosen?

The lumbar region was chosen as this region has the largest vertebrae in the spinal column and frequently undergoes minimally invasive surgeries such as corpectomy. We wanted to show that our OPF/PMMA cage can be used in this region. The cage can be used similarly in the thoracic and cervical regions as well. The following was added to the text (page 6):

“The lumbar region was chosen as this region has the largest vertebrae in the spinal column and frequently undergoes minimally invasive surgeries such as corpectomy.”

- Line 133: Please explain what is a "follower load"?

A follower load is a “load” that is added to the testing equipment to mimic the weight of the torso. This load follows the motion of the testing process by means of cables. We included a sentence (in page 7) that now reads:

“A follower load is a load that is added to the testing equipment by means of cables to mimic the weight of the torso.”

- Looks like the PMMA rod was not connected to L2 and L4. How did you determine if the cage is providing enough support or not?

The rod was not connected to the adjacent vertebral bodies as it is intended to be the filler of the hollow scaffold. During surgery, this PMMA filler is compressed by the upper weight of the body allowing it to stay in contact with the, in this specific case, L2 and L4. Comparing Cavity+Rod, where there was no PMMA, and Cavity+Rod+OPF, we can see that the PMMA rod did not improve spine’s motion during extension because the rod was not connected (during extension, and as expected in this cadaveric testing, the superior vertebral body moves away). However, the rod was able to enhance the motion during flexion in which the two vertebral bodies approach.

We modified the text (in page 11) to clarify this:

“During flexion (when the superior vertebra moves anteriorly), the L2/L4 segment moved +3.2°, which ….”.

“In extension (when the superior vertebra moves posteriorly and tends to separate from the scaffold system), however, the OPF/PMMA cage did not help improve the spine motion …”

- what's the statistic method used?

As the goal of the study was to show a proof of concept of the formulation and technique, we used a small number of samples (three samples for each formulation/group for pre- and post-expansion and one cadaveric spine for the kinematic and performing corpectomy). These small numbers did not allow us to perform any statistical analysis. This is highlighted in the imitation of the study as well

- In the abstract, the authors stated that the study "optimized the cage formulations", what was done regarding the optimization?

Thank you for the comment. We changed the concentration of different substances according to Table 1 and compared 6 different formulations to find the one that accounted for vertebral body and defect size. We removed the word optimized and changed the text to clarify this. The text now reads:

We compared six different cage formulations to account for vertebral body and defect size, and performed a cage characterization via expansion kinetics and mechanical testing evaluations.”

  1. Results
    - Line 152: Any quantitative comparison between the 6 formulations?

We did not perform any quantitative assessment to compare these formulations. The main criterion was the mechanical integrity of the samples before and after expansion. We noticed that only formulation ‘f’ was able to handle deformation without failure before and after swelling. The remaining formulations failed during expansion or upon small deformation after the expansion. We added the following statement to address this comment:

“The material of choice for the cage should be able to tolerate significant deformation so that it can be inserted at the surgical site through a small incision. Therefore, as we were developing the formulations and cages, we were qualitatively assessing the malleability and rigidity of the cages.”

- Line 157: Length, wall thickness, and internal diameter, all of them increased after expansion?

Yes! We modified the sentence (in page 8) to address this comment:

“… all of these parameters, including wall thickness, internal diameter, and length, increased about twice their original sizes.”

We also added the original and expanded dimensions (in page 5) to highlight this:

“The averaged sample dimensions were as follows: for pre-expansion, the internal diameter was 8.07mm, wall thickness was 0.78mm and length was 10mm. After expansion (100min immersion in saline), the internal diameter increased to 17.63mm, the wall thickness changed to 1.74mm, and in the length increased to 21.39mm length.”

- Figure 4: Are the curves from one specimen or averaged results?

The curves are from one cadaveric spine. Each panel in Figure 4 shows 4 motion cycles for the same scenario.

- As the authors mentioned, stiffening in lateral bending was mainly due to pedicles and screws. This again goes back to the question that how did authors determine if the cage is providing enough support or not if both ends were not attached to the vertebrates?

Thanks for this comment. As we can see from the bottom right figure in Figure 4 (lateral bending) the spine that had rods but also a cavity (blue curve) showed more motion (or less stability) than when the OPF/PMMA was added (red curve). However, we agree with the reviewer that, as the cage was not connected to the adjacent vertebral bodies, most of stability in lateral bending is given by the pedicles and screws.

  1. Discussion
    - What is the purpose of the mechanical testing on the OPF hollow cages before and after expansion? The stiffness has differences in magnitude. What was the clinical relevance of the test result?

Clinically, a cage should have small dimensions to be places in the surgical site via a small incision; the pre-expanded cage will be used to be placed in the surgical site. As the results show the pre-expanded cage has higher mechanical properties to allow for its placement without failure. While placed between the two vertebrae, it starts swelling while becoming softer. The softness of the cage at this stage allows the cage to form the shape of the empty space created by the defect, especially the both ends of the cage that form the shapes of the adjacent vertebrae without any failure. At this stage, it should be stiff enough to hold the injected polymer such as PMMA. The given formulation satisfies all these need.

- As the author mentioned, one advantage of the OPF hollow cages is that it can expand more than twice their original size during an acceptable surgical time-frame. But as described in Line 190, the procedure seems not likely to be done in a simple and fast way. Any comments?

The pre-expanded cages can be fabricated and stored in advance of surgeries. After incising and accessing a surgical site, the cage can be inserted. As soon as exposing to body’s fluid or warm saline, the cage starts expanding more than twice its original size in a timely manner.

We added the above statement to the discussion to clarify this:

“In this technique, pre-expanded cages can be fabricated and stored in advance of surgeries. After incising and accessing a surgical site, the cage can be inserted. As soon as exposing to body’s fluid, the cage starts expanding more than twice its original size in a timely manner.”

- Line 202: Please discuss what's new and unique in this study, Please summarize your most important finding in the first paragraph.

We presented a formulation for OPF that can be an alternative material for a spine cage that accounts for the size of the human vertebral bodies. We then showed using kinematic testing on a cadaveric spine that the cage increases and improves spine stability.

We have modified the first paragraph of the discussion (in page 11) to address this comment:

“In the current study, we presented six different formulations for OPF hydrogel formulations and found a formulation that can be tailored to a specific size mimicking human vertebral sizes. The OPF hollow cages expanded more than twice their original size during an acceptable surgical time-frame, becoming malleable and allowing for insertion inside a surgery site in a less invasive manner. Additionally, we performed kinematic testing and showed that the OPF/PMMA cage could provide stability to the spine while allowing for similar motions as the intact during flexion.”

- Line 228: How was it controlled to expand the cage to occupy the lesion and space created during the corpectomy?

According to our results, we know that the expansion rate is about 2.2 times the original size. Also, the defect size is known as it can be easily measured via imaging (or in cadaveric testing using a caliper/ruler). Then, the size of the dry cage can be calculated. The expanded length of the cage should be almost the same as the length of defect.

- Line 246: The statement sounds contradicting itself. If the authors believe the cage having a significant impact in flexion and lateral bending, why would the overall behavior and ROM outcomes still have a similar trend?

Thank you for the comment! We removed the contradictory statement from the text.

Reviewer 4 Report

1. The goal of this work was not achieved. No influence of size, shape, material composition of OPF/PMMA cage on spine stability was determined.

2. Cages are introduced to maintain foraminal height to prevent the nerve root from being compressed in the neuroforamina after decompression. It is well known that, in patients treated with PMMA cages, there was a significantly higher incidence of subsidence than in patients treated with PEEK cages. Why the authors decided to use the PMMA-based cage system?

3. Lines 152-153 "Formulation “f” showed the best results for the structural integrity of the cage and, thus, was chosen for the study and next steps." On what basis was this choice made?

4. Line 157 "The ratios of length, wall thickness, and internal diameter were measured for 100 minutes" This ratio is unclear. Please provide a calculation methodology.

5. Lines: 214-215 - "Our proposed OPF hollow cage, in contrast, is a biocompatible biomaterial that will degrade over time, which makes it an attractive material for spine fusion cages." On what basis it was found? - no biodegradation tests.

6. Tab. 1 Why do you choose exactly such concentrations / combinations?
7. Fig. 3 Separate scales for the same parameter are misleading.

Author Response

  1. The goal of this work was not achieved. No influence of size, shape, material composition of OPF/PMMA cage on spine stability was determined.

Thank you for the comment. The goal of the study was solely to present an OPF cage that can take into account human vertebral body and defect size and show that the OPF cage can be placed in a surgical site during corpectomy, while providing spine stability by means of kinematic evaluations. Assessing the effect of size and shape of the scaffold on the spine stability, while necessary and interesting, was not in the scope of the current study. We added a sentence (in page 13) to highlight the importance of these parameters as a limitation and future study.

“Future studies also should focus of the effect on size and shape of the cage on spine stability.”

  1. Cages are introduced to maintain foraminal height to prevent the nerve root from being compressed in the neuroforamina after decompression. It is well known that, in patients treated with PMMA cages, there was a significantly higher incidence of subsidence than in patients treated with PEEK cages. Why the authors decided to use the PMMA-based cage system?

We agree with the reviewer that PMMA spacers can contribute to higher incidence of subsidence. In our lab, we are investigating alternative polymers such as poly(propylene fumarate) (PPF) that can address several of the limitations of PMMA. The reason to use PMMA was only to show that the OPF cage works as an alternative material and surgical approach for spine surgery and delivery of cages. Additionally, we showed that our approach does provide spine stability, and would potentially improve the outcomes of minimally invasive surgeries. Instead of PMMA, other polymers, such as our PPF which allows for drug delivery and bone formation, could have been injected and tested for their efficacies.

Lines 152-153 "Formulation “f” showed the best results for the structural integrity of the cage and, thus, was chosen for the study and next steps." On what basis was this choice made?

We thank the reviewer for this comment. In order to insert the cage into a surgical site via a small incision, the pre-expanded cage should be strong enough not to break during surgery; however, it should be stiff enough to maintain its shape and not fail during injecting the polymer inside the cage once the system is placed in the spine. Therefore, the basis of choosing a formulation was the mechanical properties, or integrity, of the final product. Some of these formulations ended up failing during expansion or failed (broke) after expansion. Formulation ‘f’ showed the best outcomes; the material was soft enough to be inserted and strong enough not to fail before and after swelling.

  1. Line 157 "The ratios of length, wall thickness, and internal diameter were measured for 100 minutes" This ratio is unclear. Please provide a calculation methodology.

Thank you for the comment. We apologize for this omission. We added the original and expanded dimensions at the end of the “Fabrication and expansion of cages” section (in page 5):

“The averaged sample dimensions were as follows: for pre-expansion, internal diameter was 8.07mm, wall thickness was 0.78mm and length was 10mm. After expansion (after 100min), the internal diameter increased to 17.63mm, the wall thickness changed to 1.74mm, and in the length increased to 21.39mm length.”

  1. Lines: 214-215 - "Our proposed OPF hollow cage, in contrast, is a biocompatible biomaterial that will degrade over time, which makes it an attractive material for spine fusion cages." On what basis it was found? - no biodegradation tests.

We have added the following references to address this comment:

  1. Shin H, Temenoff JS, Mikos AG (2003) In vitro cytotoxicity of unsaturated oligo [poly (ethylene glycol) fumarate] macromers and their cross-linked hydrogels. Biomacromolecules 4 (3):552-560
  2. Gaihre B, Liu X, Lee Miller A, Yaszemski M, Lu L (2020) Poly (Caprolactone Fumarate) and Oligo [Poly (Ethylene Glycol) Fumarate]: Two Decades of Exploration in Biomedical Applications. Polymer Reviews:1-38
  1. Tab. 1 Why do you choose exactly such concentrations / combinations?

Formulation ‘a’ was the result of a previous study [1] in our group. We noticed that these concentrations failed to provide acceptable stiffness/strength for our purpose. We then changed the concentrations and fabricated new cages until we ended up formulation ‘f’. While a very small amount of this material is used to fabricate PPF, NVP is related to human health risks because of its carcinogenic characteristics. In our final formulation, we eliminated this substance that makes the final product even more biocompatible.

  1. Liu X, Paulsen A, Giambini H, Guo J, Miller AL, Lin P-C, Yaszemski MJ, Lu L (2017) A New Vertebral Body Replacement Strategy Using Expandable Polymeric Cages. Tissue Engineering Part A 23 (5-6):223-232

  2. Fig. 3 Separate scales for the same parameter are misleading.

We apologize that the reviewer finds this misleading. We have tried implementing the same scale but as the difference between the pre- and post-expansion values were remarkably large, the post-expansion values were shown almost zero. For this reason, and in order to minimize the number or redundant figures, we decided to use two different axes, highlighted with different colors, to show the differences in pre- and post-expansion outcomes. We hope this is acceptable for the reviewer.

Round 2

Reviewer 2 Report

Thank you for the prompt response and new adding.

They seem better.

One issue authors may polish. The purposes include

"first, to optimize the  cage formulations to account for vertebral body and intervertebral discs defect size, and perform a characterization via expansion kinetics and mechanical testing evaluations; second, to investigate the feasibility of the OPF/PMMA cage in providing spine stability via kinematic analyses."

After kinematic testing showed the effectiveness of this technique in providing stability to the spine. Authors conclude "the new flexible, expandable OPF formulation presented here allows developing patient-specific and sizable implants that could be placed at a surgical site using a smaller incision compared with current minimally invasive procedures.

Authors may have to rewrite this part. It seems to me that  the conclusion is not "stability". Especially the study did not include the histology evaluation. Please rewrite result summary in the section of discussion and also consistently in the abstract.

Author Response

We thank the reviewer for this comment. In the conclusion, we are referring to mechanical stability. We agree with the reviewer that other evaluations should be performed to assess stability of the construct at other levels, and we have included these in the limitation and conclusion sections. We have added the word “mechanical” to only refer to this type of stability in the text. Also we added "mechanical stability" in the abstract for consistency. We hope the reviewer agrees with our decision.

Reviewer 3 Report

Dear Authors:

Thank you for submitting the revised manuscript.

My concerns and comments about the rationale and the methodology of the biomechanical tests have been addressed.
The Method now provides more details of the study design.
Discussion about clinical relevance and the unique aspects of the current study have been added.
Comments from other reviewers have also been addressed.
Nice job!

Author Response

Thank you for your time and reviewing our manuscript.

Reviewer 4 Report

Thank you for your explanations. 

Author Response

(The authors gave the same response as above.)
